# Interest in brief resistance training workouts among older US adults with and without mobility disability

Jordan D. Kurth[1,2]*, Christopher N. Sciamanna[1,2], Margaret K. Danilovich[3], David E. Conroy[4], Kathryn H. Schmitz[5], Matthew Silvis[2], Liza S. Rovniak[2], Robert Creath[6], Ema V. Karakoleva[2]

**1** The Pennsylvania State University, University Park, Pennsylvania, United States of America, **2** Penn State College of Medicine, The Pennsylvania State University, Hershey, Pennsylvania, United States of America, **3** CJE SeniorLife, Leonard Schanfield Research Institute, Chicago, Illinois, United States of America, **4** Department of Kinesiology, The Pennsylvania State University, University Park, Pennsylvania, United States of America, **5** Department of Medicine, University of Pittsburgh, Pittsburgh, Pennsylvania, United States of America, **6** Department of Exercise Science, Lebanon Valley College, Annville, Pennsylvania, United States of America

* jdk5930@psu.edu; jkurth@pennstatehealth.psu.edu; jordandkurthphd@gmail.com

## Abstract

### Background/Objectives

Resistance training (RT) improves strength and physical function; however, only 35% of older adults meet national guidelines for muscle strengthening activities. Though time is often noted as a barrier to physical activity participation, little is known about the interest of older adults in brief RT programs. This study compared preferences for brief, more frequent vs longer, less frequent RT programs.

### Methods

A nationwide survey was conducted among 611 US adults aged 65 and over. Preference for either (1) a traditional (45 minutes per session, three days per week) RT program or (2) a short (5 minutes per session, daily) RT program was compared.

### Results

Overall, 2.2 times as many older adults preferred the daily 5-minute RT program versus the traditional RT program (68.4% v. 31.6%). Preference for the brief, daily RT program was 5.3 times higher among adults with difficulty walking (84.2% v. 15.8%) than for the traditional RT program.

### Conclusion

Preference for a daily 5-minute RT program is significantly higher than for traditional 45-minute, three times weekly programs. This difference is larger in older adults who have serious difficulty walking or climbing stairs.

which permits unrestricted use, distribution, and reproduction in any medium, provided the original author and source are credited.

**Data availability statement:** Data associated with this study is available via Penn State Data Commons at https://doi.org/10.26208/4V88-SW71.

**Funding:** The author(s) received no specific funding for this work.

**Competing interests:** One of the co-authors, Christopher Sciamanna, has an investment, such as stock, in a company which has begun to investigate the possibility of creating a business that provides exercise programs. All other authors declare that they have no known competing financial interests or personal relationships that could have appeared to influence the work reported in this paper. This does not alter our adherence to PLOS One policies on sharing data and materials.

## Significance/Implications

Brief daily RT options may help engage older adults in RT at a population level, particularly those with poorer health and mobility disability. Future investigations should evaluate differences in uptake, adherence, and outcomes from two RT programs of varying durations and frequencies.

## Introduction

One in four older adults, the fastest growing demographic group in the US [1], reports serious difficulty walking or climbing stairs, referred to as mobility disability [2]. Older adults with mobility disability have larger declines in quality of life than those with depression, anxiety or pain [3] and are 15 times more likely to develop additional activities of daily living (ADL) disabilities. In addition to health-related stress, those with mobility disability also incur an additional $10,000 each year in health care costs, including $2000 in additional out-of-pocket costs, and are 8.7 times more likely to die prematurely [4].

Systematic reviews observe that 6 months of resistance training (RT) increases strength by 50% in older adults, which improves mobility [5]. Although RT prevents and improves mobility disability [6], fewer than 20% of older adults meet national guidelines for doing RT twice per week [7], as recommended. Even when programs are free, such as Silver Sneakers, fewer than 25% participate [8] and most of those that do enroll attend less than twice per week—which is insufficient to meet national RT guidelines [9] on its own.

One approach to increase engagement in and adherence to RT is by manipulating the characteristics (i.e., duration) of the program -- making RT programs shorter. Two pieces of data support this approach. First, though older adults have more free time, as many do not work, a perceived lack of time is one of their most commonly reported barriers to RT [10,11]. Second, studies suggest that brief workouts can be effective. Systematic reviews observe that the first few sets of a strength exercise each week lead to the greatest strength gains [12]. Consistent with these findings, American College of Sports Medicine (ACSM) Guidelines note that, "A single set of resistance exercise can be effective especially among older and novice exercisers." [13].

While few studies have examined the impact of brief RT programs, studies that have examined them report positive results. Fujita and colleagues observed that a single set of repeated chair stands each day performed by older adults in a long-term care facility significantly improved knee extension torque over 12 weeks [14]. Slaughter and colleagues observed that six months of exercises consisting of only one set of repeated chair stands performed four times daily in nursing home residents led to a 12.5% reduction in time to first sit-to-stand compared to 19.1% increase in time to first sit-to-stand for controls [15]. Given the initial promise of these brief RT programs, we sought to investigate differences in the preferences for brief versus more traditional-length RT programs in older adults, particularly those for whom RT would have the largest impact (i.e., those with mobility disability).

Therefore, our objective was to understand whether a short, daily program would be preferred by older adults, especially those with mobility disability, compared to longer, three-times weekly programs. We hypothesized that the brief program would be preferred by all older adults, and that preference would be greater among those with mobility disability. Traditional exercise and RT programs for older adults that are offered through the YMCA and other fitness centers typically average 45–60 minutes three times weekly, while brief 5-minute programs are not generally offered as an exercise option for older adults. While studies note that many of the barriers to RT reported by older adults would be alleviated by brief RT sessions (e.g., fatigue), no prior studies have examined whether older adults, especially those with mobility disability, would be interested in brief RT sessions. We also conducted an exploratory qualitative analysis to clarify the reasons underlying older adults' preferences for brief vs. longer RT programs. The preliminary qualitative analyses were performed without hypotheses, to identify themes for the reasons that individuals preferred longer versus brief exercise sessions.

## Materials and methods

In April of 2023 we conducted an anonymous survey using a commercial survey company (Qualtrics) that was classified as exempt by the Office of Research Protections (Penn State University IRB# STUDY00022306). Qualtrics uses a network of survey participants from a large number of suppliers with a range of recruitment methods from across the US. Recruitment methods include: advertisements on smartphones, promotions on smartphones, referrals from lists of customers or members of organizations, social networks, smartphone games and others [16]. All participants were provided with a brief description of the study, and informed that their choosing to advance to the survey constituted agreement to participate in the study. The need for consent was waived by the institutional review board, as data was collected anonymously. To ensure data quality, surveys included (1) attention checks (i.e., factual questions with correct answers) and (2) speeding checks (i.e., eliminating responses from those who completed the online survey in less than one-third the median duration of survey completion). Quotas were placed on recruitment to ensure a representative sample of the population over 65 in the US. Qualtrics charged approximately $10 for each completed survey.

### Participants

Participants were required to be at least 65 years of age, located in the United States, and fluent in the English language. Additional target demographic quotas were placed on recruitment to ensure a sample representative of the population over 65 years of age in the US based on race, age, and sex. Target demographic quotas were placed on sex (50% male/50% female) and race/ethnicity (55% non-White Hispanic, 15% African American, 15% Asian American, 15% Alaskan Native/Native Hawaiian; 10% Hispanic). A target minimum of 556 participants (278 per preferred program) were planned to be enrolled in order to be able to detect a small (Cohen's $d = 0.2$) difference at 90% power at an $\alpha = 0.05$ within each preference group. Demographic information was collected prior to completing the rest of the survey; once quotas were met for a particular demographic combination, no additional survey completions were allowed by people in that particular demographic. Data collection occurred between April 3, 2023 and May 31, 2023.

### Instruments and measures

**Resistance training program preference.** We created a survey to understand the preferences of older adults for traditional (45 minute, three times weekly) RT sessions and brief (5 minute, daily) RT sessions. Participants were asked to select which RT program, for home use, they preferred with the following prompt: "*We are designing a strength training program for people over 65 to use at home, to improve their physical function, ability to walk and to reduce falls. Assuming that all programs include similar types of exercises, which would you prefer?*". The order of the answer options ("*5 minutes, every day*" or "*45 minutes, 3 times per week*") was presented randomly to each participant. No further information was provided about possible differences in efficacy, as the survey was designed to understand the preference for programs of

different lengths. Participants were then asked to explain their choice, with the following question: *"Why do you feel this way? Please tell us why you chose this option. Please use as much space and provide as much detail as you can."*

**Health status.** Using survey items from the National Health Interview Survey (NHIS) [17], participants were also asked to report the number of days in the past 7 days that they performed moderate intensity physical activities for at least 10 minutes at a time, vigorous intensity physical activities for at least 10 minutes at a time, walking for at least 10 minutes at a time and muscle-strengthening activities [7]. Mobility disability was assessed by using a yes/no question used in the Behavioral Risk Factor Surveillance System (BRFSS) that asked about "*serious difficulty walking or climbing stairs*".2 Falls in the past 12 months were also assessed using an item from the BRFSS "*In the past 12 months, how many times have you fallen*?", which was collapsed into yes (at least 1) or no (zero falls) [18]. Medical history was assessed using seven questions from the BRFSS, assessing presence or absence of diabetes, high cholesterol, heart disease, osteoporosis, hypertension, stroke and arthritis. Self-rated health was assessed using the single-item "In general, how is your health?" adapted from the BRFSS with responses including "excellent" (5), "very good" (4), "good" (3), "fair" (2) and "poor"(1). Smoking status was assessed using the single item adapted from the BRFSS "*Do you use a tobacco product every day*?" Use of mobility devices was assessed using self-report of using any one of three commonly-used mobility devices (cane, walker, wheelchair), using questions from the National Health and Aging Trends Study [19]. Demographic and anthropometric characteristics, such as age, gender, race, ethnicity, height and weight were assessed using items from the BRFSS and body mass index (BMI) was calculated using standard formulas.

**Analysis.** All quantitative analyses were performed in STATA 18 (College Station, TX: StataCorp LP.). The significance of differences between older adults reporting preference for brief daily RT and traditional programs were determined by Chi-squared and ANOVA tests. Exploratory qualitative analyses were performed by analyzing the first 4000 characters of text for the reasons given for participant preference, which was the free text limit for ChatGPT, first among individuals who preferred the 5-minute sessions each day and then among individuals who preferred the 45-minute sessions three times weekly. All responses provided were included in these analyses. The paid version of three AI engines were used for the analyses: ChatGPT 4.0 (OpenAI, Inc.), Copilot (Microsoft, Inc.), Perplexity (Perplexity AI, Inc). Qualitative analysis using AI technology has been shown to have adequate concurrent validity with themes identified by human coders [20,21].

## Results

### Participant profile

In April 2022, 1162 adults began the survey. Participants that did not reach the end of the survey (n = 438), whose demographic quotas were filled before they completed their survey (n = 95), that completed the survey too quickly (n = 9), that failed the attention check (n = 5), or that were identified as bots using embedded survey fields (n = 4) were removed. Therefore, the final sample size included 611 participants who completed all questions needed for this analysis. Demographics, medical history, physical function, and amount of physical activity are reported in Table 1. The average age was 72.4 years, 48% were female, 56% were white and 90% were non-Hispanic. Most participants reported hypertension and high cholesterol (62% and 57%, respectively), while fewer reported diabetes and heart disease (25% and 15%, respectively). Nearly one in six (15%) reported using a mobility device and 22% reported serious difficulty walking or climbing stairs. Approximately one-third (29%) reported falling at least once in the past year. In the previous week, participants reported an average of 4.0 days of walking, 2.9 days of moderate-intensity physical activity, 1.4 days of vigorous-intensity physical activity and 1.4 days of resistance training.

### Preferred resistance training program

Overall, 2.2 times as many older adults preferred daily 5-minute RT sessions versus traditional, three-times weekly 45-minute RT sessions (68.4% vs. 31.6%, respectively). Preferences for brief versus longer RT sessions did not differ

**Table 1. Participant characteristics by program preference.**

| | Overall (N = 611) (SD, median, range, %) | Program preference | | |
| --- | --- | --- | --- | --- |
| | | 5 min daily (SD, range or %) | 45 min 3-times/week (SD, range or %) | p-value |
| Age | 72.4 (5.1, 72, 65-90) | 72.4 (5.2, 72, 65-90) | 72.5 (4.8, 72, 65-88) | 0.76 |
| Gender, female | 294 (48.1) | 215 (73.1) | 79 (26.9) | 0.01 |
| Race, White | 341 (55.8) | 244 (71.6) | 97 (28.5) | 0.06 |
| Race, Black | 102 (16.7) | 63 (61.8) | 39 (38.2) | 0.11 |
| Ethnicity, Non Hispanic | 549 (89.8) | 374 (68.1) | 175 (31.9) | 0.65 |
| Medical History | | | | |
| BMI | 27.7 (7.0, 26.7, 8.7-58.7) | 28.3 (7.5, 27.2, 8.7-58.7) | 26.5 (5.4, 26.0, 9.6-44.8) | 0.002 |
| Smoker | 64 (10.5) | 49 (76.6) | 15 (23.4) | 0.14 |
| Diabetes* | 152 (25.2) | 113 (74.3) | 39 (25.7) | 0.08 |
| High Cholesterol* | 338 (56.5) | 235 (69.5) | 103 (30.5) | 0.56 |
| Heart disease* | 87 (14.5) | 65 (74.7) | 22 (25.3) | 0.16 |
| Osteoporosis* | 100 (17.0) | 73 (73.0) | 27 (27.0) | 0.21 |
| Hypertension* | 375 (62.1) | 266 (70.9) | 109 (29.1) | 0.11 |
| Stroke* | 40 (6.6) | 31 (77.5) | 9 (22.5) | 0.20 |
| Arthritis* | 295 (49.4) | 203 (68.8) | 92 (31.2) | 0.81 |
| Co-Morbidities (of 7*) | 2.2 (1.4, 0-7) | 2.3 (1.4, 0-6) | 2.1 (1.4, 0-7) | 0.046 |
| Self-Rated Health ** | 3.2 (0.9) | 3.1 (0.9) | 3.4 (0.9) | < 0.001 |
| Physical Function | | | | |
| Mobility device use | 94 (15.4) | 82 (87.2) | 12 (12.8) | <0.001 |
| Fell in past year | 178 (29.4) | 142 (79.8) | 36 (20.2) | <0.001 |
| Serious difficulty walking or climbing | 133 (21.8) | 112 (84.2) | 21 (15.8) | <0.001 |
| PA in Past 7 Days | | | | |
| Days of Walking | 4.0 (2.5, 0-7) | 3.8 (2.6, 0-7) | 4.6 (2.2, 0-7) | <0.001 |
| Days of Moderate PA | 2.9 (2.5, 0-7) | 2.4 (2.4, 0-7) | 3.8 (2.4, 0-7) | <0.001 |
| Days of Vigorous PA | 1.4 (2.0, 0-7) | 1.0 (1.8, 0-7) | 2.2 (2.2, 0-7) | <0.001 |
| Days of Strength Training | 1.4 (2.1, 0-7) | 1.1 (1.9, 0-7) | 2.0 (2.2, 0-7) | <0.001 |

*Note:* ** 5 = excellent, 4 = very good, 3 = good, 2 = fair, 1 = poor.

significantly by age, race or ethnicity, but women preferred the brief RT sessions more than men (p = 0.01, Table 1). Those who preferred daily 5-minute RT sessions reported performing significantly fewer days, in the past week, of walking (3.8 v. 4.6), moderate physical activity (2.4 vs. 3.8), vigorous physical activity (1.0 vs. 2.2), and days of resistance training in the past week (1.1 vs. 2.0; all p's < 0.001) Of past medical history questions, only BMI was related to preference for brief sessions, as the mean BMI was significantly higher for participants preferring the daily 5 minute RT sessions (28.3) compared to those preferring the three times weekly 45 minute RT sessions (26.5). No significant differences in preference were observed for those with and without a history of diabetes, high cholesterol, heart disease, osteoporosis, hypertension, stroke, arthritis or smoking. A significant difference was observed, however, for the total number of self-reported comorbidities (of 7) with those preferring a daily 5-minute RT program reporting 2.3 comorbidities while those preferring three times weekly 45-minute RT sessions reporting 2.1 comorbidities (p = 0.046). Participants preferring daily 5-minute RT sessions also reported worse self-rated health (3.4 versus 3.1, p < 0.001). Participants reporting any individual mobility related

difficulty (e.g., falls, mobility device use, or serious difficulty walking or climbing stairs) were significantly more likely to prefer a daily 5-minute RT session, with all comparisons being significant at a p < 0.001. For example, those reporting serious difficulty walking or climbing stairs were 5.3 times as likely to prefer a 5-minute daily RT session (84.2% v. 15.8%).

## Qualitative results

Exploratory qualitative analyses using the AI programs ChatGPT, Copilot and Perplexity can be seen in Table 2. While the AI programs differed slightly in the themes identified in their analysis, several themes were consistent: First, many older adults commented that 45 minutes of exercise is too uncomfortable or physically demanding, but 5 minutes would be possible; once participant noted that "*after about 5 minutes, my legs and my back are hurting so bad that I have to stop and rest*". Second, briefer sessions are easier to incorporate into their daily lives, as one participant noted that "*a five-minute routine would be like brushing my teeth every morning - just a regular quick task each day*". Third, some older adults believed that 5 minutes is too short to improve their health, and that 45 minutes would provide greater health benefits. noting: "*five minutes a day isn't enough time to get the heart rate up to benefit the body*" and "*I feel 45 minutes 3 times a week is much more beneficial than 5 minutes a day.*"

## Discussion

We set out to understand whether older adults preferred a brief, daily 5-minute RT program versus a standard, 45-minute three-times-per week RT program. The results showed more than twice as many older adults preferred the brief, daily RT sessions and those preferences were even stronger for those reporting serious difficulty walking or climbing stairs, where 5.3 times as many older adults preferred the brief, daily RT sessions. For most comparisons, participants preferring the daily 5-minute RT sessions reported worse health, including: worse self-reported health, a greater number of comorbidities, a higher BMI, greater use of mobility devices, greater walking difficulty, and a greater prevalence of falls in the past year.

Another key finding was the relationship between program preference and current amount of aerobic physical activity and RT. Older adults who preferred the 45-minute sessions reported 21% more days of walking, 58% more days of moderate intensity physical activity, 220% more days of vigorous intensity activity and 185% more days of RT in the previous week. These results may explain why fewer than 25% of older adults participate in traditional longer 45–60 minute exercise programs offered by fitness centers and through programs such as Silver Sneakers and Renew Active [8]. Based on these results, it seems plausible that these traditional, often longer programs based in fitness centers are attracting older adults who are more fit and capable of greater amounts of sustained exercise.

These results suggest that initial participation (i.e., uptake) may be greater if brief RT options were offered. Preliminary qualitative analyses suggest that the reasons for this preference include physical limitations that make longer sessions very difficult, and a preference for daily sessions, which may make habits easier to form. Concerns remain, among those who preferred the longer sessions, that 5-minute sessions would not be long enough to be beneficial at all, that longer sessions would be more beneficial than brief sessions, and that not having sessions each day would allow for more flexibility to schedule the sessions at times and days that work best. It remains unclear if long-term participation (i.e., adherence) would differ between short- and longer-session RT programs. However, the benefit of increased uptake of short RT programs is of significant potential public health benefit, bringing the potential for feasible RT programs to the populations that would benefit from them the most (i.e., worse self-reported health, more comorbidities, higher BMI, more use of mobility devices, more walking difficulty, and more falls in the past year). This format may also be particularly amenable to rehabilitation settings, where uptake and early adherence of exercise interventions is typically low. Again, these qualitative findings were conducted using AI tools, and are therefore to be considered exploratory and require confirmation through more traditional thematic analysis methods (e.g., manual coding or mixed-methods triangulation).

**Table 2. Themes and representative quotes from 3 artificial intelligence engines among participants preferring 5-minute daily sessions and 45 minute three times weekly sessions.**

| Theme | Representative Quotes from those preferring 5-minute daily sessions |
|---|---|
| ChatGPT | |
| Duration and Intensity of Exercise | *"45 minutes is too long to spend on exercising and I would not do it. 5 minutes a day is more doable and I would try to do it every day."* <br> *"45 minutes is too long, especially if it's strength training; thus, the only alternative was 5 minutes. The best time would be 15 minutes per day five times a week."* <br> *"I have moderate arthritis in my lumbar back & both knees, and a lumbar bone spur, which makes walking on uneven ground or using stairs difficult."* |
| Consistency and Habit Formation | *"3x a week makes me forget if I do it everyday it's just like a routine activity."* <br> *"As a retiree, I can always find five or ten minutes to perform some task that will help my balance! A five-minute routine would be like brushing my teeth every morning - just a regular quick task each day."* <br> *"Doing it every day will more than likely make it a constant daily practice that I incorporate into my life."* |
| Physical Limitations and Health Issues | *"After about 5 minutes my legs and my back are hurting so bad that I have to stop and rest."* <br> *"Because I have bad knees and it makes it difficult. Also, I work 6 days a week which gives some activity."* <br> *"I can do no more than 30 minutes exercise without resting but after a rest can return to further exercise if necessary."* |
| Co-Pilot | |
| Time Duration Preferences | *"45 minutes is too long to spend on exercising and I would not do it. 5 minutes a day is more doable and I would try to do it every day."* <br> *"45 minutes is too long. Even 15 minutes every day would be okay."* <br> *"5 minutes a day would fit better in my daily life and be more sustainable in the long run."* |
| Physical Limitations and Health Conditions | *"After about 5 minutes, my legs and my back are hurting so bad that I have to stop and rest."* <br> *"At this time, I cannot walk for 45 minutes. In the past, walking was my next favorite form of exercise behind weightlifting."* <br> *"Bad knees."* |
| Ease of Commitment and Routine Integration | *"5 minutes a day would be ideal for me. I KNOW I would do it, and making 5 minutes available every day I can do."* <br> *"A little exercise goes a long way. I can only do it a little bit at a time."* <br> *"As a retiree, I can always find five or ten minutes to perform some task that will help my balance! A five-minute routine would be like brushing my teeth every morning - just a regular quick task each day. Sign me up!"* |
| Perplexity | |
| Preference for shorter duration due to physical limitations | *"5 minutes a day would be easy to do"* <br> *"5 minutes is achievable. It's tough to take time to do 45 minutes, let alone 3 times."* <br> *"Although five minutes is not a lot of time, at this point and time in my life, 45 is too much."* |
| Easier to incorporate daily routine into existing schedule | *"Doing it every day will more than likely make it a constant daily practice that I incorporate into my life."* <br> *"Easier to get into habit doing daily. 5min means not having to schedule the activity."* <br> *"Five minutes a day would be more convenient for me. I wouldn't be out of breath for that short length of time."* |
| Consistency and habit formation | *"Consistency is better than volume. Consistent exercise seems more beneficial."* <br> *"Doing every day doing exercise in small amounts for me is what I'm used to doing. my body can withstand that. Doing it every day gets one into the habit of doing it."* <br> *"Every day for a short time is ok with me. every day would be more likely to make it a habit"* |
| **Theme** | **Representative Quotes from those preferring 45 minute three-time weekly sessions** |
| ChatGPT | |
| Adequacy of Exercise Duration | *"5 minutes a day really is not enough."* <br> *"A strength training program requires commitment, 5 minutes is not enough time to do a worthwhile routine."* <br> *"First of all, five minutes a day isn't enough time to get the heart rate up to benefit the body. Forty-five minutes, on the other hand, is and more beneficial, in my opinion."* |
| Flexibility and Scheduling | *"3 times a week makes it more flexible to find time between other responsibilities."* <br> *"45 minutes 3X per week is better than 5 minutes everyday because 45 minutes is just enough for the body to build up strength."* <br> *"Does not tie up every day.* |
| Physical and Mental Benefits | *"Invigorates both the mind and body."* <br> *"45 minutes is more beneficial."* <br> *"Hard work one day with a one day rest or moderate level of exercise."* |

*(Continued)*

**Table 2.** (Continued)

| Theme | Representative Quotes from those preferring 5-minute daily sessions |
|---|---|
| Co-Pilot | |
| Adequacy of Exercise Duration | "45 minutes 3X per week is better than 5 minutes everyday because 45 minutes is just enough for the body to build up strength."<br>"45 minutes can be a good workout and worth the time to change into workout clothes. I would feel like I accomplished more."<br>"I believe that to gain strength and cardio fitness a longer sustained time is necessary." |
| Flexibility and Scheduling | "3 times a week makes it more flexible to find time between other responsibilities."<br>"It would be easier to do it 3 days a week than to do it everyday. I get lots of exercise at work."<br>"Work hard 3 days and do less rigorous activity the other days." |
| Physical Limitations | "I am recovering from cancer and I am trying to build my strength back up and possible play some golf."<br>"I have moderate arthritis in my lumbar back & both knees, and a lumbar bone spur, which makes walking on uneven ground or using stairs difficult."<br>"I am in constant pain and any physical activity is excruciatingly painful." |
| Perplexity | |
| Adequacy of Exercise Duration | "45 minutes every other day seems like the right amount of time"<br>"45 minutes is more beneficial. Invigorates both the mind and body."<br>"45 minutes three times a week, will give individuals more of a consistency in their exercise program." |
| 5 minutes is too short | "5 minutes a day really is not enough"<br>"5 minutes is too little, even daily. 5 minutes is too short"<br>"5 minutes just isn't enough" |
| Longer workouts are more beneficial | "I feel 45 minutes 3 times a week is much more beneficial than 5 minutes a day."<br>"I feel that 5 minutes a day won't benefit me as much as 45 minutes 3 times a week"<br>"I think the longer workouts would be more beneficial" |

While the study was conducted with a national sample in a commonly used survey panel, and recruited a diverse sample that was representative of the US population over the age of 65 without requiring in-person data collection, the conclusions must be judged with several limitations in mind. First, the sample was of individuals with Internet access while only 75% of older adults report access to the Internet [22]. A nationwide survey using the same Qualtrics panel in 2020 observed greater rates of college completion (55.8%) than observed by the Census Bureau (37.9%) [23], suggesting that the sampling method used by Qualtrics may be biased towards respondents that are more educated than the general public [22]. This may be related to why the rates of serious difficulty walking or climbing stairs in this study (22%) were lower than the percentage of adults 65 and older reporting difficulty "walking or climbing stairs" (26.9%) in a national study in 2016 [2]. Second, the survey only asked about preference, not intent, specific plans for change, or actual exercise behavior, so it remains unknown whether these stated preferences would lead more older adults to actually participate in brief programs if offered, or continue them longer if they began. In a previous study, more than 50% of older adults expressed an interest in a free strength training program [24], yet fewer than 25% of older adults with access to free programs that include strength training actually participate [8]. There is also a well-established intention-behavior gap that has not been addressed in these findings [25].Third, we chose 45-minutes as a comparison for the 5-minute program, though there are many exercise programs for older adults, and their length varies between 30 minutes for the Otago program [26] to 90 minutes for the Fit & Strong program [27]. Our results would likely have been different had we chosen a different comparator duration or a range of duration options. Our findings do not suggest that a 5-minute daily strength training workout is the most preferred possible strength training format for older adults. Fourth, while it was not measured, the wording in the survey did not include any information about the comparative effectiveness of the RT programs. The question may have, therefore, given participants the impression that both programs were both effective, and possibly equally so, when in reality the 45-minute program would likely be more effective, assuming (importantly) that adherence to the two programs was also equal [28]. This survey was done to understand the interest of older adults in brief RT sessions, assuming they could

be designed to improve outcomes, so the findings may only represent an interest in having access to effective brief RT sessions. Fifth, the qualitative analysis was preliminary, as using AI for qualitative analysis is relatively new, so that human coders may have reached different conclusions. Lennon and colleagues observed a close correlation between themes identified by human coders and AI programs (Cohen's kappa 0.62−.72) [21], yet few studies have examined the accuracy of AI to code qualitative data [20]. The themes identified by the three different programs, however, were quite similar, which should increase confidence that future, formal qualitative analyses would identify similar themes.

In conclusion, these results, from a large nationwide sample of older adults, suggest that older adults would prefer a briefer daily RT option rather than traditional 45-minute, three days per week RT programs, currently being offered in the US as part of Medicare Advantage or Medicare Supplemental insurance [29]. Preference for the brief program was even stronger for older adults that would benefit most (e.g., those with mobility disability, who are less active, who have higher BMI). These results suggest that if brief, equally effective programs were offered, uptake and adherence to those programs may be better than traditional, longer, programs. Future studies should examine both adherence and outcome impacts of short RT programs compared to longer, traditional RT programs [30].

## Supporting information

**S1 File. Strobe checklist.**
(DOCX)

## Author contributions

**Conceptualization:** Jordan D. Kurth, Christopher N. Sciamanna, Margaret K Danilovich, David E. Conroy, Kathryn H. Schmitz, Matthew Silvis, Liza S Rovniak.

**Data curation:** Jordan D. Kurth, Christopher N. Sciamanna, Robert Creath, Ema V. Karakoleva.

**Formal analysis:** Jordan D. Kurth, Christopher N. Sciamanna.

**Investigation:** Jordan D. Kurth, Liza S Rovniak.

**Methodology:** Jordan D. Kurth, Christopher N. Sciamanna, Margaret K Danilovich, David E. Conroy, Kathryn H. Schmitz.

**Project administration:** Jordan D. Kurth, Christopher N. Sciamanna.

**Supervision:** Jordan D. Kurth.

**Validation:** Jordan D. Kurth.

**Writing – original draft:** Jordan D. Kurth, Christopher N. Sciamanna, Margaret K Danilovich, David E. Conroy, Kathryn H. Schmitz, Matthew Silvis, Liza S Rovniak, Robert Creath, Ema V. Karakoleva.

**Writing – review & editing:** Jordan D. Kurth, Christopher N. Sciamanna, Margaret K Danilovich, David E. Conroy, Kathryn H. Schmitz, Matthew Silvis, Liza S Rovniak, Robert Creath, Ema V. Karakoleva.

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
