## [Editor Report · Decision Letter 0]

24 Feb 2025

PONE-D-25-05366Interest in brief resistance training workouts among older US adults with and without mobility disabilityPLOS ONE

Dear Dr. Kurth,

Thank you for submitting your manuscript to PLOS ONE. After careful consideration, we feel that it has merit but does not fully meet PLOS ONE’s publication criteria as it currently stands. Therefore, we invite you to submit a revised version of the manuscript that addresses the points raised during the review process.

We look forward to receiving your revised manuscript.

Kind regards,

Ateya Megahed Ibrahim El-eglany

Academic Editor

PLOS ONE

Journal Requirements:

2. Please provide additional details regarding participant consent. In the ethics statement in the Methods and online submission information, please ensure that you have specified (1) whether consent was informed and (2) what type you obtained (for instance, written or verbal, and if verbal, how it was documented and witnessed). If your study included minors, state whether you obtained consent from parents or guardians. If the need for consent was waived by the ethics committee, please include this information

3. Thank you for stating the following in the Competing Interests section: “Christopher Sciamanna has an investment, such as stock, in a company which has begun to investigate the possibility of creating a business that provides exercise programs. All other authors declare that they have no known competing financial interests or personal relationships that could have appeared to influence the work reported in this paper.”

4. We noted in your submission details that a portion of your manuscript may have been presented or published elsewhere. [Data related to this publication have been published previously in PLOS One. The published manuscript has been provided with this submission.] Please clarify whether this publication was peer-reviewed and formally published. If this work was previously peer-reviewed and published, in the cover letter please provide the reason that this work does not constitute dual publication and should be included in the current manuscript.

Additional Editor Comments:

Editorial Recommendation

Before we can proceed with your manuscript for further processing, we request that you:

Strengthen the discussion by acknowledging the limitations of self-reported preferences versus actual exercise behavior and their implications for generalizability.

Provide more details on sample representativeness and potential biases, including how quota sampling was enforced and addressing self-selection bias due to digital survey recruitment.

Discuss the feasibility and long-term sustainability of a 5-minute RT program, considering factors such as program adherence, supervision, and real-world implementation.

Clarify how missing data were handled in the analysis and whether any sensitivity analyses were conducted to assess potential bias.

Address the lack of blinding in survey responses and discuss any measures taken to minimize social desirability bias.

Enhance the discussion on the practical implications of shorter RT programs, particularly for public health interventions and rehabilitation settings.

Consider adding a follow-up study examining actual adherence, physical function outcomes, and health benefits of a 5-minute RT program compared to traditional RT regimens.

We appreciate your contributions to this field and look forward to your revised manuscript. Please submit your revisions along with a point-by-point response addressing these concerns.

---

## [Author Response · Author response to Decision Letter 1]

4 Mar 2025

Journal Requirements:

Thank you, and sorry for the citation formatting oversight. I believe that the style requirements are now met.

2. Please provide additional details regarding participant consent. In the ethics statement in the Methods and online submission information, please ensure that you have specified (1) whether consent was informed and (2) what type you obtained (for instance, written or verbal, and if verbal, how it was documented and witnessed). If your study included minors, state whether you obtained consent from parents or guardians. If the need for consent was waived by the ethics committee, please include this information

Sorry for this oversight. The following statement has been added to the Methods section of the manuscript:

All participants were provided with a brief description of the study, and informed that their choosing to advance to the survey constituted agreement to participate in the study. The need for written consent was waived by the institutional review board, as data was collected anonymously.

3. Thank you for stating the following in the Competing Interests section: “Christopher Sciamanna has an investment, such as stock, in a company which has begun to investigate the possibility of creating a business that provides exercise programs. All other authors declare that they have no known competing financial interests or personal relationships that could have appeared to influence the work reported in this paper.”

Thank you. The following statement has been added to the cover letter:

One of the co-authors, Christopher Sciamanna, has an investment, such as stock, in a company which has begun to investigate the possibility of creating a business that provides exercise programs. All other authors declare that they have no known competing financial interests or personal relationships that could have appeared to influence the work reported in this paper. This does not alter our adherence to PLOS One policies on sharing data and materials.

4. We noted in your submission details that a portion of your manuscript may have been presented or published elsewhere. [Data related to this publication have been published previously in PLOS One. The published manuscript has been provided with this submission.] Please clarify whether this publication was peer-reviewed and formally published. If this work was previously peer-reviewed and published, in the cover letter please provide the reason that this work does not constitute dual publication and should be included in the current manuscript.

The following paragraph is now included in the cover letter:

This research is related to a manuscript previously published in PLOS One. The related manuscript is focused on evaluating the equivalence of preferences and self-efficacy ratings, as well as identifying the possible implications of using either preference or self-efficacy ratings to design strength training programs for older adults to enhance long-term maintenance. In contrast, this research identifies patterns of health issues/comorbidities and predictors that influence strength training program preference. Therefore, this work does not constitute dual publication. The related manuscript published in PLOS One has been included with this submission.

Thank you for highlighting this. We have an established methodology of hosting data for availability related to publications within Penn State University. The timeline for this process is typically a matter of days.

Thank you. This change has been made.

Editorial Recommendation

Strengthen the discussion by acknowledging the limitations of self-reported preferences versus actual exercise behavior and their implications for generalizability.

Good point. The following statement is now included in the discussion section:

Second, the survey only asked about preference, not intent, specific plans for change, or actual exercise behavior, so it remains unknown whether these stated preferences would lead more older adults to actually participate in shorter programs if offered, or continue them longer if they began. In a previous study, more than 50% of older adults expressed an interest in a free strength training program, yet fewer than 25% of older adults with access to free programs that include strength training actually participate.

Provide more details on sample representativeness and potential biases, including how quota sampling was enforced and addressing self-selection bias due to digital survey recruitment.

Thank you for highlighting this issue. The following has been added in the participants section:

Demographic information was collected prior to completing the rest of the survey; once quotas were met for a particular demographic combination, no additional survey completions were allowed by people in that particular demographic.

The paragraph below also appears in the discussion section:

While the study was conducted with a national sample in a commonly used survey panel, and recruited a diverse sample that was representative of the US population over the age of 65, the conclusions must be judged with several limitations in mind. First, the sample was of individuals with Internet access while only 75% of older adults report access to the Internet. A nationwide survey using the same Qualtrics panel in 2020 observed greater rates of college completion (55.8%) than observed by the Census Bureau (37.9%), suggesting that the sampling method used by Qualtrics may be biased towards respondents that are more educated than the general public.

Discuss the feasibility and long-term sustainability of a 5-minute RT program, considering factors such as program adherence, supervision, and real-world implementation.

Thank you for this suggestion. The followings section has been added to the discussion section:

It remains unclear if long-term participation (i.e., adherence) would differ between short- and longer-session RT programs. However, the benefit of increased uptake of short RT programs is of significant potential public health benefit, bringing the potential for feasible RT programs to the populations that would benefit from them the most (i.e., worse self-reported health, more comorbidities, higher BMI, more use of mobility devices, more walking difficulty, and more falls in the past year). This format may also be particularly amenable to rehabilitation settings, where uptake and early adherence of exercise interventions is typically low.

Clarify how missing data were handled in the analysis and whether any sensitivity analyses were conducted to assess potential bias.

Thank you for this clarification. There was not missing data in this analysis, as part of the Qualtrics quota system requires a completed survey to meet the quota numbers. This has been clarified in the participant profile section:

In April 2022, 1162 adults began the survey. Participants that did not reach the end of the survey (n = 438), whose demographic quotas were filled before they completed their survey (n = 95), that completed the survey too quickly (n = 9), that failed the attention check (n = 5), or that were identified as bots using embedded survey fields (n = 4) were removed. Therefore, the final sample size included 611 participants who completed all questions needed for this analysis.

Address the lack of blinding in survey responses and discuss any measures taken to minimize social desirability bias.

Thank you for raising this issue. The survey responses were not conditional by participant, so there was no blinding that could occur.

The following has been added to the discussion section to reflect actions taken to minimize social desirability bias:

While the study was conducted with a national sample in a commonly used survey panel, and recruited a diverse sample that was representative of the US population over the age of 65 without requiring in-person data collection, the conclusions must be judged with several limitations in mind.

Enhance the discussion on the practical implications of shorter RT programs, particularly for public health interventions and rehabilitation settings.

Thank you for this suggestion. The followings section has been added to the discussion section:

It remains unclear if long-term participation (i.e., adherence) would differ between short- and longer-session RT programs. However, the benefit of increased uptake of short RT programs is of significant potential public health benefit, bringing the potential for feasible RT programs to the populations that would benefit from them the most (i.e., worse self-reported health, more comorbidities, higher BMI, more use of mobility devices, more walking difficulty, and more falls in the past year).

Consider adding a follow-up study examining actual adherence, physical function outcomes, and health benefits of a 5-minute RT program compared to traditional RT regimens.

We could not agree more. We are currently conducting a long-term randomized controlled trial of exactly this. The following statement has been added to the discussion.

Future studies should examine both adherence and outcome impacts of short RT programs compared to longer, traditional RT programs.

We appreciate your contributions to this field and look forward to your revised manuscript. Please submit your revisions along with a point-by-point response addressing these concerns.

Thank you very much for your time and effort to improve this manuscript. We appreciate your perspectives and insight.

---

## [Decision Letter · Decision Letter 1]

2 Jul 2025

PONE-D-25-05366R1Interest in brief resistance training workouts among older US adults with and without mobility disabilityPLOS ONE

Dear Dr. Jordan D. Kurth

Thank you for submitting your manuscript to PLOS ONE. After careful consideration, we feel that it has merit but does not fully meet PLOS ONE’s publication criteria as it currently stands. Therefore, we invite you to submit a revised version of the manuscript that addresses the points raised during the review process.

We look forward to receiving your revised manuscript.

Kind regards,

Mehrnaz Kajbafvala, Ph.D

Academic Editor

PLOS ONE

Reviewers' comments:

Reviewer's Responses to Questions

**Comments to the Author**

1. If the authors have adequately addressed your comments raised in a previous round of review and you feel that this manuscript is now acceptable for publication, you may indicate that here to bypass the “Comments to the Author” section, enter your conflict of interest statement in the “Confidential to Editor” section, and submit your "Accept" recommendation.

Reviewer #1: (No Response)

Reviewer #2: (No Response)

2. Is the manuscript technically sound, and do the data support the conclusions?

Reviewer #1: No

Reviewer #2: Yes

3. Has the statistical analysis been performed appropriately and rigorously? 

Reviewer #1: Yes

Reviewer #2: Yes

4. Have the authors made all data underlying the findings in their manuscript fully available?

Reviewer #1: Yes

Reviewer #2: Yes

5. Is the manuscript presented in an intelligible fashion and written in standard English?

Reviewer #1: Yes

Reviewer #2: Yes

6. Review Comments to the Author

Reviewer #1: I am pleased to review the project titled “Interest in brief resistance training workouts among older US adults with and without mobility disability.” The research topic is an important one regarding the evaluation of barriers to physical activity participation. However, although the study is interesting and the findings likely benefit future researchers, it needs further revisions.

Major Issues.

1. A subjective comparison was made between “a traditional (45 minutes per session, three days per week) RT program” versus “a short (5 minutes per session, daily) RT program.” However, “No further information was provided about possible differences in 11 efficacy” (page 5, line 10). Unless you provide further information to participants, how will one choose one over the other? I believe, if I offer a one-minute daily program more participants would prefer that one over a 5-minute program, unless we provide them information regarding the benefits of the program. For example, it has been found that the effect sizes obtained in mind-body practice are similar to those observed in RCTs testing the efficacy of cholinesterase inhibitors, such as donepezil, in participants with a similar cognitive status indicating mind-body practices’ potential to be an alternative and complementary practice (see https://doi.org/10.1016/j.archger.2020.104319).

2. Why are there only two options? Why not a program in between? Further, the supporting evidences (Fujita and colleagues, Slaughter and colleagues; page 3, lines 21-25) were conducted in long-term care and on more vulnerable participants. Therefore, how they are equivalent in community-dwelling older adults. Needs more explanation.

3. The authors mentioned “Participants were required to be at least 65 years of age, a US citizen…” (page 4, line 19). They also mentioned “Participants were required to be at least 65 years of age, located in the United States, and fluent in the English 24 language” (page 4, line 23). What is the difference between US citizens and those located in the US, from the study’s perspective? From a cultural perspective, it differs a lot. Why the participants need to be fluent in English? Why not with adequate communication skills in English? Again, it matters for older adults who are not US citizens. This needs more clarification.

4. What theoretical bases the authors used to build this study?

Reviewer #2: General Comments

This manuscript presents the results of a national online survey investigating preferences among older adults for either brief (5-minute daily) or traditional (45-minute, thrice-weekly) resistance training programs. The topic is highly relevant, as engagement in resistance training remains very low in older populations despite its well-established health benefits—particularly for those with mobility impairments.

The paper is well-written, clearly structured, and addresses a practical question with important implications for the design of inclusive and scalable exercise interventions for older adults. The sample is large and diverse, and the inclusion of mobility-impaired participants strengthens the relevance of the findings for at-risk groups.

The authors are to be commended for taking a pragmatic, behaviorally informed approach. However, I have a few comments and suggestions that I believe would further strengthen the manuscript.

Major Comments

- About the use of AI for qualitative analysis:

The manuscript uses AI tools (ChatGPT, Copilot, Perplexity) for preliminary analysis of open-ended survey responses. While innovative, this approach is relatively novel and still under evaluation in qualitative research. The manuscript does acknowledge this, but I suggest a more cautious tone in the discussion. Readers would benefit from a clearer statement that the qualitative findings are exploratory and require confirmation through more traditional thematic analysis methods (e.g., manual coding or mixed-methods triangulation).

- Assumption of effectivity between RT programs:

The survey presents the two RT options as equally effective, but in reality, the clinical outcomes of 5-minute versus 45-minute sessions are not equivalent based on current evidence. While I understand the purpose of isolating preference, I recommend that this be more clearly explained as a methodological limitation. Readers unfamiliar with RT literature might misinterpret the comparison as implying equivalency in training benefit.

- Sampling:

The study acknowledges the bias introduced by using an online survey platform, but it would be helpful to contextualize this further. For example, does the education level or health status of this sample differ from national averages for older adults? The claim of “representativeness” should be nuanced to reflect potential selection bias (e.g., more digitally literate or health-aware individuals).

- From preference to behavior

A key limitation is that the study assesses preference, not actual intent or adherence. While this is addressed in the discussion, I encourage the authors to expand on the gap between stated preferences and real-world behavior. Citing behavioral intention models (e.g., Theory of Planned Behavior) might help reinforce this point.

Minor Comments

- Please clarify how qualitative responses were selected and whether any pre-processing or filtering occurred before analysis.

- Table 1: Consider adding a row indicating education level or income if available, as these are known to influence both health behaviors and survey response biases.

- Occasionally the terms “brief RT program” and “shorter program” are used interchangeably. For clarity, it may help to standardize this language across the manuscript.

- All citations appear appropriate and current. Good use of evidence to support background and rationale.

Conclusion and Recommendation:

This manuscript addresses an important gap in the literature with practical relevance to the design of RT programs for older adults. The findings are timely, well-supported, and clearly communicated. With minor revisions to improve transparency around the qualitative methods and the assumptions made in the preference comparison, I believe this paper will make a valuable contribution.

7. PLOS authors have the option to publish the peer review history of their article (what does this mean? ). If published, this will include your full peer review and any attached files.

**Do you want your identity to be public for this peer review?** For information about this choice, including consent withdrawal, please see our Privacy Policy .

Reviewer #1: **Yes: ** Kallol Kumar Bhattacharyya

Reviewer #2: **Yes: ** Eduardo Carballeira

---

## [Author Response · Author response to Decision Letter 2]

3 Jul 2025

Reviewer #1

I am pleased to review the project titled “Interest in brief resistance training workouts among older US adults with and without mobility disability.” The research topic is an important one regarding the evaluation of barriers to physical activity participation. However, although the study is interesting and the findings likely benefit future researchers, it needs further revisions.

Thank you for your time and effort to improve this manuscript.

1. A subjective comparison was made between “a traditional (45 minutes per session, three days per week) RT program” versus “a short (5 minutes per session, daily) RT program.” However, “No further information was provided about possible differences in 11 efficacy” (page 5, line 10). Unless you provide further information to participants, how will one choose one over the other? I believe, if I offer a one-minute daily program more participants would prefer that one over a 5-minute program, unless we provide them information regarding the benefits of the program. For example, it has been found that the effect sizes obtained in mind-body practice are similar to those observed in RCTs testing the efficacy of cholinesterase inhibitors, such as donepezil, in participants with a similar cognitive status indicating mind-body practices’ potential to be an alternative and complementary practice (see https://doi.org/10.1016/j.archger.2020.104319).

This is a good question. We intentionally limited the information provided to participants, as we were only interested in their preference for the frequency and duration of exercise programs (as in the second half of the quoted sentence, “No further information was provided about possible differences in efficacy, as the survey was designed to understand the preference for programs of different lengths.” [emphasis added]. When choosing to participate in an exercise program in the real-world, the frequency and duration of the activity is the primary information that is provided (along with the type of activity, which in this case is held constant between the two options).

It is possible that preferences may be greater for other, even shorter programs. Our aim was to investigate preferences for brief programs generally, as represented by 5-minutes in this case, against the vast majority of exercise program formats, which are 45-60 minutes, 3 times per week.

These limitations stemming from the provided choices, and their implications in the conclusions reached, are mentioned in the discussion section:

“Third, we chose 45-minutes as a comparison for the 5-minute program, though there are many exercise programs for older adults, and their length varies between 30 minutes for the Otago program [25] to 90 minutes for the Fit & Strong program [26]. Our results would likely have been different had we chosen a different comparator duration or a range of duration options. Our findings do not suggest that a 5-minute daily strength training workout is the most preferred possible strength training format for older adults. Fourth, while it was not measured, the wording in the survey did not include any information about the comparative effectiveness of the RT programs. The question may have, therefore, given participants the impression that both programs were both effective, and possibly equally so despite the well-described dose-effect of RT among individuals who adhere to protocols [27]. This survey was done to understand the interest of older adults in shorter RT sessions, assuming they could be designed to improve outcomes, so the findings may only represent an interest in having access to effective brief RT sessions.”

To better reflect this preference as not one of shorter vs. longer, but a singular brief option vs. standard of care, the term “brief” has replaced “shorter” throughout the manuscript.

2. Why are there only two options? Why not a program in between? Further, the supporting evidences (Fujita and colleagues, Slaughter and colleagues; page 3, lines 21-25) were conducted in long-term care and on more vulnerable participants. Therefore, how they are equivalent in community-dwelling older adults. Needs more explanation.

As mentioned above, our aim was not to identify the most preferred exercise duration (thus requiring several options); our goal was to determine preferences between two options: the standard format, and a brief, daily format. To clarify this further, the following statement has been added to the discussion section:

“Our findings do not suggest that a 5-minute daily strength training workout is the most preferred possible strength training format for older adults.”

Additionally, as noted above, to better reflect this preference as not one of shorter vs. longer, but a singular brief option vs. standard of care, the term “brief” has replaced “shorter” throughout the manuscript.

The supporting evidence mentioned is in more vulnerable populations than general, community-dwelling older adults. We do not suggest that this is evidence that these programs are equally effective in community-dwelling older adults. Instead, we only suggest that there is initial promise that they may be effective in community-dwelling older adults, especially those that are very de-conditioned, and that it warrants further investigation.

We feel that the preliminary evidence in vulnerable populations, coupled with the ACSM guidelines statement provided (“A single set of resistance exercise can be effective especially among older and novice exercisers.”) are sufficient justification to investigate interest in brief exercise programs for community-dwelling older adults.

3. The authors mentioned “Participants were required to be at least 65 years of age, a US citizen…” (page 4, line 19). They also mentioned “Participants were required to be at least 65 years of age, located in the United States, and fluent in the English 24 language” (page 4, line 23). What is the difference between US citizens and those located in the US, from the study’s perspective? From a cultural perspective, it differs a lot. Why the participants need to be fluent in English? Why not with adequate communication skills in English? Again, it matters for older adults who are not US citizens. This needs more clarification.

Thank you for catching this issue. The correct stipulation is that participants need be located in the United States. This was less a function of the research question, and instead a requirement given the sampling methods used. Citizenship was not verified nor considered as a part of eligibility criteria. All participants in the Qualtrics panels must reside in the United States. This has been clarified in-text.

We also did not differentiate between fluency and adequate communication skills. If a participant was comfortable participating in the study (which was only offered in English), they were allowed to participate.

4. What theoretical bases the authors used to build this study?

The study is driven by empirical data instead of theory. Only 20% of older adults perform sufficient amounts of resistance training. The most commonly cited barrier to participating is not having enough time. Evidence suggests that brief exercise sessions can still be effective. Therefore, we wanted to investigate the interest in an exercise program that was brief, thereby potentially addressing a major barrier to participation, while maintaining the possibility of the program being effective.

Reviewer #2: General Comments

This manuscript presents the results of a national online survey investigating preferences among older adults for either brief (5-minute daily) or traditional (45-minute, thrice-weekly) resistance training programs. The topic is highly relevant, as engagement in resistance training remains very low in older populations despite its well-established health benefits—particularly for those with mobility impairments.

The paper is well-written, clearly structured, and addresses a practical question with important implications for the design of inclusive and scalable exercise interventions for older adults. The sample is large and diverse, and the inclusion of mobility-impaired participants strengthens the relevance of the findings for at-risk groups.

The authors are to be commended for taking a pragmatic, behaviorally informed approach. However, I have a few comments and suggestions that I believe would further strengthen the manuscript.

Thank you for your time and effort to review and improve this manuscript.

- About the use of AI for qualitative analysis:

The manuscript uses AI tools (ChatGPT, Copilot, Perplexity) for preliminary analysis of open-ended survey responses. While innovative, this approach is relatively novel and still under evaluation in qualitative research. The manuscript does acknowledge this, but I suggest a more cautious tone in the discussion. Readers would benefit from a clearer statement that the qualitative findings are exploratory and require confirmation through more traditional thematic analysis methods (e.g., manual coding or mixed-methods triangulation).

Thank you for this suggestion. We agree, and have added the following statement to the end of the paragraph discussing the qualitative findings in the discussion:

“Again, these qualitative findings were conducted using AI tools, and are therefore to be considered exploratory and require confirmation through more traditional thematic analysis methods (e.g., manual coding or mixed-methods triangulation).”

- Assumption of effectivity between RT programs:

The survey presents the two RT options as equally effective, but in reality, the clinical outcomes of 5-minute versus 45-minute sessions are not equivalent based on current evidence. While I understand the purpose of isolating preference, I recommend that this be more clearly explained as a methodological limitation. Readers unfamiliar with RT literature might misinterpret the comparison as implying equivalency in training benefit.

Thank you for this suggestion. We agree, and have amended the statement in the discussion section to read as below:

“Fourth, while it was not measured, the wording in the survey did not include any information about the comparative effectiveness of the RT programs. The question may have, therefore, given participants the impression that both programs were both effective, and possibly equally so, when in reality the 45-minute program would likely be more effective, assuming (importantly) that adherence to the two programs was also equal [27].”

- Sampling:

The study acknowledges the bias introduced by using an online survey platform, but it would be helpful to contextualize this further. For example, does the education level or health status of this sample differ from national averages for older adults? The claim of “representativeness” should be nuanced to reflect potential selection bias (e.g., more digitally literate or health-aware individuals).

Thank you for requesting this clarification. The qualifier “representative based on race, ethnicity, and sex” [emphasis added] has been added to the statement in the methods section.

The additional biases of the sampling methods are mentioned in detail in the discussion section. While this information is not available for this specific sample, it stands to reason (and is conservative to assume) that the known biases of this sampling method also apply here:

“First, the sample was of individuals with Internet access while only 75% of older adults report access to the Internet [22]. A nationwide survey using the same Qualtrics panel in 2020 observed greater rates of college completion (55.8%) than observed by the Census Bureau (37.9%) [23], suggesting that the sampling method used by Qualtrics may be biased towards respondents that are more educated than the general public [22]. This may be related to why the rates of serious difficulty walking or climbing stairs in this study (22%) were lower than the percentage of adults 65 and older reporting difficulty “walking or climbing stairs” (26.9%) in a national study in 2016 [2].”

- From preference to behavior

A key limitation is that the study assesses preference, not actual intent or adherence. While this is addressed in the discussion, I encourage the authors to expand on the gap between stated preferences and real-world behavior. Citing behavioral intention models (e.g., Theory of Planned Behavior) might help reinforce this point.

Thank you. We have expanded this section to mention the intention-behavior gap:

“There is also a well-established intention-behavior gap that has not been addressed in these findings [25].”

Minor Comments

- Please clarify how qualitative responses were selected and whether any pre-processing or filtering occurred before analysis.

Thank you. We have clarified this in the methods section:

“Exploratory qualitative analyses were performed by analyzing the first 4000 characters of text for the reasons given for participant preference, which was the free text limit for ChatGPT, first among individuals who preferred the 5-minute sessions each day and then among individuals who preferred the 45-minute sessions three times weekly. All responses provided were included in these analyses.”

- Table 1: Consider adding a row indicating education level or income if available, as these are known to influence both health behaviors and survey response biases.

Unfortunately these variables were not available. However, we conservatively assume this sample has a higher than average income and education level based on previous samples using similar methods.

- Occasionally the terms “brief RT program” and “shorter program” are used interchangeably. For clarity, it may help to standardize this language across the manuscript.

Thank you for this suggestion. The term has been changed to “brief” throughout the manuscript.

- All citations appear appropriate and current. Good use of evidence to support background and rationale.

Thank you.

Conclusion and Recommendation:

This manuscript addresses an important gap in the literature with practical relevance to the design of RT programs for older adults. The findings are timely, well-supported, and clearly communicated. With minor revisions to improve transparency around the qualitative methods and the assumptions made in the preference comparison, I believe this paper will make a valuable contribution.

Thank you again for your time and effort to improve this manuscript.

---

## [Decision Letter · Decision Letter 2]

21 Jul 2025

Interest in brief resistance training workouts among older US adults with and without mobility disability

PONE-D-25-05366R2

Dear Dr. Jordan D. Kurth,

We’re pleased to inform you that your manuscript has been judged scientifically suitable for publication and will be formally accepted for publication once it meets all outstanding technical requirements.

Kind regards,

Mehrnaz Kajbafvala, Ph.D

Academic Editor

PLOS ONE

**Comments to the Author**

1. If the authors have adequately addressed your comments raised in a previous round of review and you feel that this manuscript is now acceptable for publication, you may indicate that here to bypass the “Comments to the Author” section, enter your conflict of interest statement in the “Confidential to Editor” section, and submit your "Accept" recommendation.

Reviewer #1: All comments have been addressed

Reviewer #2: All comments have been addressed

2. Is the manuscript technically sound, and do the data support the conclusions?

Reviewer #1: Yes

Reviewer #2: Yes

3. Has the statistical analysis been performed appropriately and rigorously? 

Reviewer #1: Yes

Reviewer #2: Yes

4. Have the authors made all data underlying the findings in their manuscript fully available?

Reviewer #1: Yes

Reviewer #2: Yes

5. Is the manuscript presented in an intelligible fashion and written in standard English?

Reviewer #1: Yes

Reviewer #2: Yes

6. Review Comments to the Author

Reviewer #1: Comments to the Author

I found the revised version of the manuscript, “Interest in brief resistance training workouts among older US adults with and without mobility disability,” much stronger in its purpose; I am happy with the author's responses to the reviewers' comments. All the best!

Reviewer #2: All changes made by the authors are adequate and addressed the concerns of the reviewer. Therefore, I consider that the paper, in its current version, meets the necessary requirements to be published in PLOS One.

7. PLOS authors have the option to publish the peer review history of their article (what does this mean? ). If published, this will include your full peer review and any attached files.

**Do you want your identity to be public for this peer review?** For information about this choice, including consent withdrawal, please see our Privacy Policy .

Reviewer #1: **Yes: ** Kallol Kumar Bhattacharyya

Reviewer #2: **Yes: ** Eduardo Carballeira

---

## [Editor Report · Acceptance letter]

PONE-D-25-05366R2

PLOS ONE

Dear Dr. Kurth,

I'm pleased to inform you that your manuscript has been deemed suitable for publication in PLOS ONE. Congratulations! Your manuscript is now being handed over to our production team.

Kind regards,

on behalf of

Dr. Mehrnaz Kajbafvala

Academic Editor

PLOS ONE